# On the Convergence and Calibration of Deep Learning with Differential Privacy

**Zhiqi Bu**                                                                      *zbu@upenn.edu*
*University of Pennsylvania*

**Hua Wang**                                                                   *wanghua@upenn.edu*
*University of Pennsylvania*

**Zongyu Dai**                                                               *daizy@sas.upenn.edu*
*University of Pennsylvania*

**Qi Long**                                                                      *qlong@upenn.edu*
*University of Pennsylvania*

**Reviewed on OpenReview:** *https://openreview.net/forum?id=K0CAGgjYS1*

## Abstract

Differentially private (DP) training preserves the data privacy usually at the cost of slower convergence (and thus lower accuracy), as well as more severe mis-calibration than its non-private counterpart. To analyze the convergence of DP training, we formulate a continuous time analysis through the lens of neural tangent kernel (NTK), which characterizes the per-sample gradient clipping and the noise addition in DP training, for arbitrary network architectures and loss functions. Interestingly, we show that the noise addition only affects the privacy risk but not the convergence or calibration, whereas the per-sample gradient clipping (under both flat and layerwise clipping styles) only affects the convergence and calibration.

Furthermore, we observe that while DP models trained with small clipping norm usually achieve the best accurate, but are poorly calibrated and thus unreliable. In sharp contrast, DP models trained with large clipping norm enjoy the same privacy guarantee and similar accuracy, but are significantly more *calibrated*. Our code can be found at `https://github.com/woodyx218/opacus_global_clipping`.

## 1 Introduction

Deep learning has achieved tremendous success in many applications that involve crowdsourced information, e.g., face image, emails, financial status, and medical records. However, using such sensitive data raises severe privacy concerns on a range of image recognition, natural language processing and other tasks (Cadwalladr & Graham-Harrison, 2018; Rocher et al., 2019; Ohm, 2009; De Montjoye et al., 2013; 2015). For a concrete example, researches have recently demonstrated multiple successful privacy attacks on deep learning models, in which the attackers can re-identify a member in the dataset using the location or the purchase record, via the membership inference attack (Shokri et al., 2017; Carlini et al., 2019). In another example, the attackers can extract a person's name, email address, phone number, and physical address from the billion-parameter GPT-2 (Radford et al., 2019) via the extraction attack (Carlini et al., 2020). Therefore, many studies have applied differential privacy (DP) (Dwork et al., 2006; Dwork, 2008; Dwork et al., 2014; Mironov, 2017; Duchi et al., 2013; Dong et al., 2019), a mathematically rigorous approach, to protect against leakage of private information (Abadi et al., 2016; McSherry & Talwar, 2007; McMahan et al., 2017; Geyer et al., 2017). To achieve this gold standard of privacy guarantee, since the seminal work (Abadi et al., 2016), DP optimizers (including DP-SGD/Adam (Abadi et al., 2016; Bassily et al., 2014; Bu et al., 2019), DP-SGLD (Wang et al.,

2015; Li et al., 2019; Zhang et al., 2021), DP-FedSGD and DP-FedAvg (McMahan et al., 2017)) are applied to train the neural networks while preserving high accuracy for prediction.

Algorithmically speaking, DP optimizers have two extra steps in comparison to the non-DP standard optimizers: the per-sample gradient clipping and the random noise addition, so that DP optimizers descend in the direction of the clipped, noisy, and averaged gradient (see Equation (4.1)). These extra steps protect the resulting models against privacy attacks via the Gaussian mechanism (Dwork et al., 2014, Theorem A.1), at the expense of an empirical performance degradation compared to the non-DP deep learning, in terms of much slower convergence and lower utility. For example, state-of-the-art CIFAR10 accuracy with DP is $\approx 70\%$ without pre-training (Papernot et al., 2020) (while non-DP networks can easily achieve over 95% accuracy) and similar performance drops have been observed on facial images, tweets, and many other datasets (Bagdasaryan et al., 2019; Kurakin et al., 2022).

Empirically, many works have evaluated the effects of noise scale, batch size, clipping norm, learning rate, and network architecture on the privacy-accuracy trade-off (Abadi et al., 2016; Papernot et al., 2020). However, despite the prevalent usage of DP optimizers, little is known about its convergence behavior from a theoretical viewpoint, which is necessary to understand and improve the deep learning with differential privacy.

We notice some previous attempts by (Chen et al., 2020; Bu et al., 2022; Song et al., 2021; Bu et al., 2022), which either analyze the DP-SGD in the convex setting or rely on extra assumptions in the deep learning setting.

**Our Contributions** In this work, we establish a principled framework to analyze the dynamics of DP deep learning, which helps demystify the phenomenon of the privacy-accuracy trade-off.

- We explicitly characterize the *general training dynamics* of deep learning with DP-GD in Fact 4.1. We show a fundamental influence of the DP training on the NTK matrix, which causes the convergence to worsen.

- This characterization leads to the *convergence analysis* for DP training with small or large clipping norm, in Theorem 1 and Theorem 2, respectively.

- We demonstrate via numerous experiments that a small clipping norm generally leads to more accurate but less calibrated DP models, whereas a large clipping norm effectively mitigates the *calibration* issue, preserves a similar accuracy, and provides the *same privacy guarantee*.

- We conduct the first experiments on DP and calibration with large models at the Transformer level.

To elaborate on the notion of *calibration* (Guo et al., 2017; Niculescu-Mizil & Caruana, 2005), a critical performance measure besides accuracy and privacy, we provide a concrete example as follow. A classifier is calibrated if its average accuracy, over all samples it predicts with $p$ confidence (the probability assigned on its output class), is close to $p$ $(0 < p < 1)$. That is, a calibrated classifier's predicted confidence matches its accuracy. We observe that DP models using a small clipping norm are oftentimes too over-confident to be reliable (the predicted confidence is much higher than the actual accuracy), while a large clipping norm is amazingly effective on mitigating the mis-calibration.

## 2 Background

### 2.1 Differential privacy notion

We provide the definition of DP (Dwork et al., 2006; 2014) as follows.

**Definition 2.1.** A randomized algorithm $M$ is $(\varepsilon, \delta)$-differentially private (DP) if for any neighboring datasets $S, S'$ differ by an arbitrary sample, and for any event $E$,

$$\mathbb{P}[M(S) \in E] \leqslant e^{\varepsilon} \mathbb{P}\left[M\left(S'\right) \in E\right] + \delta. \tag{2.1}$$

Given a deterministic function $G(S)$, adding noise proportional to $G$'s sensitivity makes it private. This is known as the Gaussian mechanism, as stated in Lemma 2.2 and widely used in DP deep learning.

**Lemma 2.2** (Theorem A.1 (Dwork et al., 2014); Theorem 2.7 (Dong et al., 2019))**.** *Define the $\ell_2$ sensitivity of any function $G$ to be $R := \sup_{S,S'} \|G(S) - G(S')\|_2$ where the supreme is over all neighboring datasets$(S, S')$. Then the Gaussian mechanism $\hat{G}(S) = G(S) + \sigma R \cdot \mathcal{N}(0, \mathbf{I})$ is $(\epsilon, \delta)$-DP for some $\epsilon$ depending on $(\sigma, p, \delta)$, where $p$ is the sampling ratio (e.g. batch size / total sample size).*

We note that the interdependence among $\epsilon$ and $(\sigma, n, p, \delta)$ can be characterized by various privacy accountants, including Moments accountant (Abadi et al., 2016; Canonne et al., 2020), Gaussian differential privacy (GDP) (Dong et al., 2019; Bu et al., 2019), Fourier accountant (Koskela et al., 2020), Edgeworth Accountant (Wang et al., 2022), etc., each based on a different composition theory that accumulates the privacy risk $\epsilon(\sigma, n, p, \delta, T)$ differently over $T$ iterations.

### 2.2 Deep learning with differential privacy

DP deep learning (Google; Facebook) uses a general optimizer, e.g. SGD and Adam, to update the neural networks with the

$$\text{privatized gradient: } \sum_i C_i(R) \cdot \frac{\partial \ell_i}{\partial \mathbf{w}} + \sigma R \cdot \mathcal{N}(0, \mathbf{I}), \tag{2.2}$$

where $\mathbf{w}$ is the trainable parameters of the network, $\frac{\partial \ell_i}{\partial \mathbf{w}}$ is the i-th per-sample gradient of loss $\ell$, and $\sigma$ is the noise scale that determines the privacy risk. Specifically, $C_i(R)$ is the clipping factor with the clipping norm $R$, which restricts the norm of the clipped gradient in that $\|C_i(R)\frac{\partial \ell_i}{\partial \mathbf{w}}\| \leq R$. There are multiple ways to design such an clipping factor. The most generic clipping (Abadi et al., 2016) uses $C_i = \min\{1, R/\|\frac{\partial \ell_i}{\partial \mathbf{w}}\|\}$, the automatic clipping (Bu et al., 2022) uses $C_i = 1/(\|\frac{\partial \ell_i}{\partial \mathbf{w}}\| + 0.01)$ or the normalization $C_i = 1/\|\frac{\partial \ell_i}{\partial \mathbf{w}}\|$, and the global clipping uses $C_i = \mathbb{I}\{\frac{\partial \ell_i}{\partial \mathbf{w}} \leq R\}$ to be defined in Appendix D. In this work, we focus on the traditional clipping (Abadi et al., 2016) and observe that

$$\text{clipping/normalization} \Longleftrightarrow R/\left\|\frac{\partial \ell_i}{\partial \mathbf{w}}\right\| \overset{\text{small}R}{\Longleftarrow} C_i = \min\left\{1, R/\left\|\frac{\partial \ell_i}{\partial \mathbf{w}}\right\|\right\} \overset{\text{large}R}{\Longrightarrow} C_i = 1 \Longleftrightarrow \text{no clipping.}$$

In equation 2.2, the privatized gradient has two unique components compared to the standard non-DP gradient: the per-sample gradient clipping (to bound the sensitivity of the gradient) and the random noise addition (to guarantee the privacy of models). Empirical observations have found that optimizers with the per-sample gradient clipping, even when no noise is present, have much worse accuracy at the end of training (Abadi et al., 2016; Bagdasaryan et al., 2019). On the other hand, noise addition (without the per-sample clipping), though slows down the convergence, can lead to comparable or even better accuracy at the convergence (Neelakantan et al., 2015). Therefore, it is important to characterize the effects of the clipping and the noising, which are under-studied while widely-applied in DP deep learning.

## 3 Warmup: Convergence of Non-Private Gradient Method

We start by reviewing the standard non-DP Gradient Descent (GD) for ***arbitrary neural network*** and ***arbitrary loss***. In particular, we analyze the training dynamics of a neural network using the neural tangent kernel (NTK) matrix[1].

Suppose a neural network $f$[2] is governed by weights $\mathbf{w}$, with samples $\boldsymbol{x}_i$ and labels $y_i$ $(i = 1, ..., n)$. Denote the prediction by $f_i = f(\boldsymbol{x}_i, \mathbf{w})$, and the per-sample loss by $\ell_i = \ell(f(\boldsymbol{x}_i, \mathbf{w}), y_i)$, whereas the optimization

---

[1]We emphasize that our analysis are not limited to the infinitely wide or over-parameterized neural networks. Put differently, we don't assume the NTK matrix $\mathbf{H}$ to be deterministic nor nearly time-independent, as was the case in (Arora et al., 2019a; Lee et al., 2019; Du et al., 2018; Allen-Zhu et al., 2019; Zou et al., 2020; Fort et al., 2020; Arora et al., 2019b).

[2]The neural network $f$ (and thus the loss $\ell$ and $L$) is assumed to be differentiable following the convention of existing literature (Du et al., 2018; Allen-Zhu et al., 2019; Xie et al., 2020; Bu et al., 2021b), in the sense that sub-gradient exists everywhere. This differentiability is a necessary foundation of the back-propagation for deep learning.

loss $L$ is the average of per-sample losses,

$$L(\mathbf{w}) = \frac{1}{n}\sum_{i=1}^{n}\ell(f(\boldsymbol{x}_i,\mathbf{w}),y_i).$$

In discrete time, the gradient descent with a learning rate $\eta$ can be written as:

$$\mathbf{w}(k+1) = \mathbf{w}(k) - \eta\frac{\partial L}{\partial\mathbf{w}}^{\top} = \mathbf{w}(k) - \frac{\eta}{n}\sum_i\frac{\partial\ell_i}{\partial\mathbf{w}(k)}.$$

In continuous time, the corresponding *gradient flow*, i.e., the ordinary differential equation (ODE) describing the weight updates with an infinitely small learning rate $\eta \to 0$, is then:

$$\dot{\mathbf{w}}(t) = -\frac{\partial L}{\partial\mathbf{w}(t)}^{\top} = -\frac{1}{n}\sum_i\frac{\partial\ell_i}{\partial\mathbf{w}(t)}.$$

Applying the chain rules to the gradient flow, we obtain the following general dynamics of the loss $L$,

$$\dot{L} = \frac{\partial L}{\partial\mathbf{w}}\dot{\mathbf{w}} = -\frac{\partial L}{\partial\mathbf{w}}\frac{\partial L}{\partial\mathbf{w}}^{\top} = -\frac{\partial L}{\partial\boldsymbol{f}}\frac{\partial\boldsymbol{f}}{\partial\mathbf{w}}\frac{\partial\boldsymbol{f}}{\partial\mathbf{w}}^{\top}\frac{\partial L}{\partial\boldsymbol{f}}^{\top} = -\frac{\partial L}{\partial\boldsymbol{f}}\mathbf{H}(t)\frac{\partial L}{\partial\boldsymbol{f}}^{\top}, \tag{3.1}$$

where $\frac{\partial L}{\partial\boldsymbol{f}} = \frac{1}{n}(\frac{\partial\ell_1}{\partial f_1},...,\frac{\partial\ell_n}{\partial f_1}) \in \mathbb{R}^{1\times n}$, and the Gram matrix $\mathbf{H}(t) := \frac{\partial\boldsymbol{f}}{\partial\mathbf{w}}\frac{\partial\boldsymbol{f}}{\partial\mathbf{w}}^{\top} \in \mathbb{R}^{n\times n}$ is known as the NTK matrix, which is positive semi-definite and crucial to analyzing the convergence behavior.

To give a concrete example, let $\ell$ be the MSE loss $\ell_i(\mathbf{w}) = (f(\boldsymbol{x}_i,\mathbf{w}) - y_i)^2$ and $L_{\text{MSE}} = \frac{1}{n}\sum_i\ell_i(\mathbf{w}) = \frac{1}{n}\sum_i(f_i - y_i)^2$, then $\dot{L}_{\text{MSE}} = -4(\boldsymbol{f} - \boldsymbol{y})^{\top}\mathbf{H}(t)(\boldsymbol{f} - \boldsymbol{y})/n^2$. Furthermore, if $\mathbf{H}(t)$ is positive definite, the MSE loss $L_{\text{MSE}} \to 0$ exponentially fast (Du et al., 2018; Allen-Zhu et al., 2019; Zou et al., 2020) , and the cross-entropy loss $L_{\text{CE}} \to 0$ at rate $O(1/t)$ (Allen-Zhu et al., 2019).

## 4 Continuous-time Convergence of DP Gradient Descent

In this section, we analyze the weight dynamics and loss dynamics of DP-GD with an arbitrary clipping function in *continuous-time* analysis. That is, we study only the gradient flow of the training dynamics as the learning rate $\eta$ tends to 0. Our analysis can generalize to other optimizers such as DP-SGD, DP-HeavyBall, and DP-Adam.

### 4.1 Effect of Noise Addition on Convergence

Our first result is simple yet surprising: the *gradient flow* of a stochastic noisy GD with non-zero noise equation 4.1 is the same as that of the *gradient flow* without the noise in equation 4.2. Put it differently, the noise addition has no effect on the convergence of DP optimizers in the *limit of continuous time analysis*. We note that DP-GD shares some similarity to another noisy gradient method, known as the stochastic gradient Langevin dynamics (SGLD Welling & Teh (2011)). However, while DP-GD has a noise magnitude proportional to $\eta$ and thus corresponds to a deterministic gradient flow, SGLD has a noise magnitude proportional to $\sqrt{\eta}$, which is much larger when we let $\eta \to 0$ in the limit, and thus corresponds to a different continuous-time behavior: its gradient flow is a stochastic differential equation driven by a Brownian motion. We will extend this comparison to the discrete time in Section 4.5.

To elaborate this point, we consider the DP-GD with Gaussian noise, following the notation in equation 2.2,

$$\mathbf{w}(k+1) = \mathbf{w}(k) - \frac{\eta}{n}\left(\sum_i C_i\frac{\partial\ell_i}{\partial\mathbf{w}(k)} + \sigma R \cdot \mathcal{N}(0,\mathbf{I})\right). \tag{4.1}$$

Notice that this general dynamics covers both the standard non-DP GD ($\sigma = 0$ and, $C_i = 1$ if no clipping, or $C_i = c$ if batch clipping) and DP-GD with any clipping function. Through Fact 4.1 (see proof in Appendix B), we claim that the gradient flow of equation 4.1 is the same ODE (not SDE) regardless of the value of $\sigma$. That is, different $\sigma$ always results in the same gradient flow as $\eta/n \to 0$.

**Fact 4.1.** For all $\sigma \geq 0$, the gradient descent in equation 4.1 corresponds to the continuous gradient flow

$$\dot{\mathbf{w}}(t) = -\frac{1}{n}\sum_i \frac{\partial \ell_i}{\partial \mathbf{w}(t)}C_i(t). \tag{4.2}$$

This result indeed aligns the conventional wisdom[3] of tuning the clipping norm $C$ first (e.g. setting $\sigma = 0$ or small) then the noise scale $\sigma$, since the convergence is more sensitive to the clipping. We validate Fact 4.1 in Figure 1 by experimenting on CIFAR10 with small learning rate.

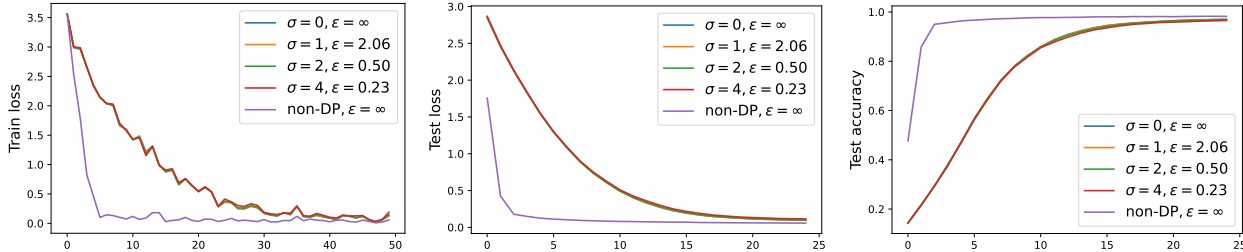

Figure 1: For fixed $R = 1, \eta = 0.1$, ViT-base trained with DP-SGD under various noise $\sigma$ has similar performance on CIFAR10 (setting in Section 5.3). Here 'non-DP' means both $\sigma = 0$ and no clipping. Notice that the loss curves for different $\sigma$ are very similar (though not the same) to each other, because we fix the random seed at the beginning of each iteration among different runs. This is to eliminate the potential difference from uncontrolled random realizations for fair comparison.

## 4.2 Effect of Per-Sample Clipping on Convergence

We move on to analyze the effect of the per-sample clipping on the DP training equation 4.2. It has been empirically observed that the per-sample clipping results in worse convergence and accuracy even without the noise (Bagdasaryan et al., 2019). We highlight that the NTK matrix is the key to understanding the convergence behavior. Specifically, the per-sample clipping affects NTK through its linear algebra properties, especially the positive semi-definiteness, which we define below in two notions for a *general* matrix.

**Definition 4.2.** For a (not necessarily symmetric) matrix $A$, it is

1. *positive in quadratic form* if and only if $\boldsymbol{x}^\top A \boldsymbol{x} \geq 0$ for every non-zero $\boldsymbol{x}$;

2. *positive in eigenvalues* if and only if all eigenvalues of $A$ are non-negative.

These two positivity definitions are equivalent for a symmetric or Hermitian matrix, but not so for non-symmetric matrices. We illustrate this difference in Appendix A with some concrete examples. Next, we introduce two styles of per-sample clippings, both can work with any clipping function.

**Flat clipping style.** The DP-GD described in equation 4.1, with the gradient flow equation 4.2, is equipped with the *flat* clipping (McMahan et al., 2018). In words, the flat clipping upper bounds the entire gradient vector by a norm $R$. Using the chain rules, we get

$$\dot{L} = \frac{\partial L}{\partial \mathbf{w}}\dot{\mathbf{w}} = -\frac{1}{n^2}\sum_j \frac{\partial \ell_j}{\partial \mathbf{w}}\sum_i \frac{\partial \ell_i}{\partial \mathbf{w}}C_i = -\frac{\partial L}{\partial \boldsymbol{f}}\mathbf{H}\mathbf{C}\frac{\partial L}{\partial \boldsymbol{f}}^\top, \tag{4.3}$$

where $\mathbf{C}(t) = \text{diag}(C_1, \cdots, C_n)$ and $C_i(t)$ is defined in Section 2.2.

---

[3]See `github.com/pytorch/opacus/blob/master/tutorials/building_image_classifier.ipynb` and Section 3.3 in (Kurakin et al., 2022).

**Layerwise clipping style.** We additionally analyze another per-sample clipping style – the *layerwise clipping* (Abadi et al., 2016; McMahan et al., 2017; Phan et al., 2017). Unlike the flat clipping, the layerwise clipping upper bounds the $r$-th layer's gradient vector by a layer-dependent norm $R_r$. Therefore, the DP-GD and its gradient flow with this layerwise clipping are:

$$\mathbf{w}_r(k+1) = \mathbf{w}_r(k) - \frac{\eta}{n}\left(\sum_i \frac{\partial \ell_i}{\partial \mathbf{w}_r} C_{i,r} + \sigma R_r \cdot \mathcal{N}(0,1)\right) \quad \text{and} \quad \dot{\mathbf{w}}_r(t) = -\frac{1}{n}\sum_i \frac{\partial \ell_i}{\partial \mathbf{w}_r} C_{i,r}.$$

Then the loss dynamics is obtained by the chain rules:

$$\dot{L} = \sum_r \frac{\partial L}{\partial \mathbf{w}_r}\dot{\mathbf{w}}_r = -\frac{\partial L}{\partial \boldsymbol{f}}\sum_r \mathbf{H}_r \mathbf{C}_r \frac{\partial L}{\partial \boldsymbol{f}}^\top, \tag{4.4}$$

where the layerwise NTK matrix $\mathbf{H}_r = \frac{\partial \boldsymbol{f}}{\partial \mathbf{w}_r}\frac{\partial \boldsymbol{f}}{\partial \mathbf{w}_r}^\top$, and $\mathbf{C}_r(t) = \text{diag}(C_{1,r}, \cdots, C_{n,r})$.

In short, from equation 4.3 and equation 4.4, the per-sample clipping precisely changes the NTK matrix from $\mathbf{H} \equiv \sum_r \mathbf{H}_r$, in the standard non-DP training, to $\mathbf{HC}$ in DP training with flat clipping, and to $\sum_r \mathbf{H}_r \mathbf{C}_r$ in DP training with layerwise clipping. Subsequently, we will show that this may break the NTK's positivity and harm the convergence of DP training.

### 4.3 Small Per-Sample Clipping Norm Breaks NTK Positivity

We show that the small clipping norm $R$ breaks the positive semi-definiteness of the NTK matrix[4].

**Theorem 1.** *For an arbitrary neural network and a loss convex in $f$, suppose at least some per-sample gradients are clipped ($\exists i, C_i < 1$) in the gradient flow of DP-GD, and assume $\mathbf{H}(t) \succ 0$, then:*

1. *for flat clipping style, the loss dynamics is equation 4.3 and the NTK matrix is $\mathbf{H}(t)\mathbf{C}(t)$, which may not be symmetric nor positive in quadratic form, but is positive in eigenvalues.*

2. *for layerwise clipping style, the loss dynamics is equation 4.4 and the NTK matrix is $\sum_r \mathbf{H}_r(t)\mathbf{C}_r(t)$, which may not be symmetric nor positive in quadratic form or in eigenvalues.*

3. *for both flat and layerwise clipping styles, the loss $L(t)$ may not decrease monotonically.*

4. *if the loss $L(t)$ converges with $\dot{L}(t) \to 0$[5], for the flat clipping style, it converges to 0; for the layerwise clipping style, it may converge to a non-zero value.*

We prove Theorem 1 in Appendix B, which states that the symmetry of NTK is almost surely broken by the clipping using small clipping norm. If furthermore the positive definiteness of NTK is broken, then severe issues may arise in the loss convergence, which is depicted in Figure 1 and Figure 8.

### 4.4 Large Per-Sample Clipping Norm Preserves NTK Positivity

Now we switch gears to large clipping norm $R$. Suppose at each iteration, $R$ is sufficiently large so that no per-sample gradient is clipped ($C_i = 1$), i.e. the per-sample clipping is not effective. Thus, the gradient flow of DP-GD is the same as that of non-DP GD. Hence we obtain the following result.

**Theorem 2.** *For an arbitrary neural network and a loss convex in $f$, suppose none of the per-sample gradients are clipped ($\forall i, C_i = 1$) in the gradient flow of DP-GD, and assuming $\mathbf{H}(t) \succ 0$, then:*

1. *for both flat and layerwise clipping styles, the loss dynamics is equation 3.1 and the NTK matrix is $\mathbf{H}(t)$, which is symmetric and positive definite.*

---

[4]It is a fact that the product of a symmetric and positive definite matrices and a positive diagonal matrix may not be symmetric nor positive in quadratic form. This is shown in Appendix A.

[5]Note that it is possible that $L(t)$ converges yet $\dot{L}(t) \not\to 0$, e.g. when uniform convergence is not satisfied.

2. *if the loss $L(t)$ converges with $\dot{L}(t) \to 0$, for both flat and layerwise clipping styles, the loss $L(t)$ decreases monotonically to 0.*

We prove Theorem 2 in Appendix B and the benefits of large clipping norm are assessed in Section 5.2. Our findings from Theorem 1 and Theorem 2 are visualized in the left plot of Figure 10 and summarized in Table 1.

| Clipping type | NTK matrix | Symmetric NTK | Positive in quadratic form | Positive in eigenvalues | Loss convergence | Monotone loss decay | To zero loss |
|---|---|---|---|---|---|---|---|
| No clipping | $\mathbf{H} \equiv \sum_r \mathbf{H}_r$ | ✓ | ✓ | ✓ | ✓ | ✓ | ✓ |
| Batch clipping | $c\mathbf{H} \equiv c\sum_r \mathbf{H}_r$ | ✓ | ✓ | ✓ | ✓ | ✓ | ✓ |
| Large $R$ clipping (Flat & layerwise) | $\mathbf{H} \equiv \sum_r \mathbf{H}_r$ | ✓ | ✓ | ✓ | ✓ | ✓ | ✓ |
| Small $R$ clipping (Flat) | $\mathbf{H}\mathbf{C}$ | ✗ | ✗ | ✓ | ✗ | ✗ | ✓ |
| Small $R$ clipping (Layerwise) | $\sum_r \mathbf{H}_r\mathbf{C}_r$ | ✗ | ✗ | ✗ | ✗ | ✗ | ✗ |

Table 1: Effects of per-sample gradient clipping on gradient flow. Here "Yes/No" means guaranteed or not and the loss refers to the training set. "Loss convergence" is conditioned on $\mathbf{H}(t) \succ 0$.

### 4.5 Connection to Bayesian Deep Learning

When $R$ is sufficiently large, all per-sample gradients are not clipped ($C_i = 1, \forall i$), and DP-SGD is essentially the SGD with independent Gaussian noise. This is indeed the SGLD (with a different learning rate) that is commonly used to train Bayesian neural networks.

$$\text{DP-SGD:} \quad \mathbf{w}(k+1) - \mathbf{w}(k) = -\frac{\eta_{\text{DP-SGD}}}{B}\left(\sum_i \frac{\partial l_i}{\partial \mathbf{w}} + \sigma R \cdot \mathcal{N}(0, I)\right),$$

$$\text{SGLD:} \quad \mathbf{w}(k+1) - \mathbf{w}(k) = -\frac{\eta_{\text{SGLD}}n}{2B}\left(\sum_i \frac{\partial l_i}{\partial \mathbf{w}}\right) + \sqrt{\eta_{\text{SGLD}}}\mathcal{N}(0, I),$$

where $n$ is the total number of samples and $B$ is mini-batch size. Clearly, DP-SGD (with the right combination of hyperparameters) is a special form of SGLD by setting $\eta_{\text{DP-SGD}} = \eta_{\text{SGLD}}n/2$ and $\sigma R\frac{n}{2B} = 1/\sqrt{\eta_{\text{SGLD}}}$.

Similarly, DP-HeavyBall with large $R$ can be viewed as stochastic gradient Hamiltonian Monte Carlo. This equivalence relation opens new doors to understanding DP optimizers by borrowing the rich literature from the Bayesian learning. Especially, the uncertainty quantification of Bayesian neural network implies the amazing calibration of large-$R$ DP optimization in Section 5.2.

## 5 Discrete-time DP Optimization: privacy, accuracy, calibration

Now, we focus on the more practical analysis when the learning rate $\eta$ is not infinitely small, i.e. the *discrete time* analysis. In this regime, the gradient flow in equation 4.2 may deviate from the dynamics of the actual training, especially when the added noise is not small, e.g. when the privacy budget $\epsilon$ is stringent and thus requires a large $\sigma$.

Nevertheless, state-of-the-art DP accuracy can be achieved under settings that is well-approximated by our gradient flow. For example, large pre-trained models such as GPT2 (0.8 billion parameters) (Bu et al., 2022; Li et al., 2021) and ViT (0.3 billion parameters) (Bu et al., a) are typically trained using small learning rates around 0.0001. In addition, the best DP models are trained with large batch size $n$, e.g. (Li et al., 2021) have used a batch size 6000 to train RoBERTa on MNLI and QQP datasets, and (Kurakin et al., 2022; De et al., 2022; Mehta et al., 2022) have used batch sizes $n$ from $10^4$ to $10^6$, i.e. full batch, to achieve state-of-the-art DP accuracy on ImageNet. These settings all result in very small noise magnitude $\eta\sigma R/n$ in the optimization[6], so that the noise has small effects on the accuracy (and the calibration), as illustrated in Figure 1. Consequently, we focus on only analyzing the effect of different clipping norms $R$.

---

[6]Here the noise magnitude discussed is per parameter. It is empirically verified that the total noise magnitude for models with millions of parameters can be also small or even dimension-independent when the gradients are low-rank (Li et al., 2022).

### 5.1 Privacy analysis

From Lemma 2.2, we highlight that DP optimizers with all clipping norms *have the same privacy guarantee*, independent of the choice of the privacy accountant, because the privacy risk $\epsilon$ only depends on the noise scale $\sigma$ (i.e. the noise-to-sensitivity ratio). We summarize this common fact in Fact 5.1, which motivates the ablation study on $R$ in most literature of DP deep learning. Consequently, one can use a larger clipping norm that benefits the calibration, while remaining equally DP as using a smaller clipping norm.

**Fact 5.1** (Abadi et al. (2016)). DP optimizers with the same noise scale $\sigma$ are equally $(\epsilon(\sigma), \delta(\sigma))$-DP, independent of the choice of the clipping norm $R$.

*Proof of Fact 5.1.* Firstly, we show that the privatized gradient in equation 2.2 has a privacy guarantee that only depends on $\sigma$, not $R$, regardless of which privacy accountant is adopted. This can be seen because (1) the sum of per-sample clipped gradient $\sum_i C_i \boldsymbol{g}_t^{(i)}$ has a sensitivity of $\max_{i \in B_t} \|C_i \boldsymbol{g}_t^{(i)}\| \leq R$ by the triangular inequality, and (2) the noise $\sigma R$ is proportional to $R$ and hence fixing the noise-to-signal ratio at $\sigma R/R = \sigma$, regardless of the choice of $R$. Therefore, the privacy guarantee is the same and independent of $R$. Secondly, it is well-known that the post-processing of a DP mechanism is equally DP, thus any optimizer (e.g. SGD or Adam) that leverages the same privatized gradient in equation 2.2 has the same DP guarantee. $\square$

### 5.2 Accuracy and Calibration

In the following sections, we reveal a novel phenomenon that DP optimizers play important roles in producing well-calibrated and reliable models.

In $M$-class classification problems, we denote the probability prediction for the $i$-th sample as $\boldsymbol{\pi}_i \in \mathbb{R}^M$ so that $f(\boldsymbol{x}_i) = \text{argmax}(\boldsymbol{\pi}_i)$, then the accuracy is $\mathbf{1}\{f(\boldsymbol{x}_i) = y_i\}$. The confidence, i.e., the probability associated with the predicted class, is $\hat{P}_i := \max_{k=1}^{M}[\boldsymbol{\pi}_i]_k$ and a good calibration means the confidence is close to the accuracy[7]. Formally, we employ three popular calibration metrics from (Naeini et al., 2015): the test loss, i.e. the negative log-likelihood (NLL), the Expected Calibration Error (ECE), and the Maximum Calibration Error (MCE).

$$\text{ECE:} \quad \mathbb{E}_{\hat{P}_i}\left[\left|\mathbb{P}(f(\boldsymbol{x}_i) = y_i|\hat{P}_i = p) - p\right|\right], \qquad \text{MCE:} \quad \max_{p \in [0,1]}\left|\mathbb{P}(f(\boldsymbol{x}_i) = y_i|\hat{P}_i = p) - p\right|.$$

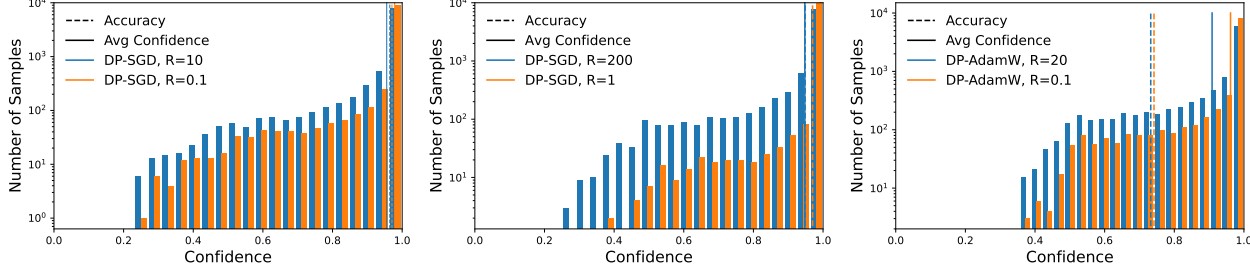

Figure 2: Confidence histograms on CIFAR 10 (left), MNIST (middle), and SNLI (right).

| | ECE % | | | MCE % | | |
|---|---|---|---|---|---|---|
| | non-DP | DP (small $R$) | DP (large $R$) | non-DP | DP (small $R$) | DP (large $R$) |
| CIFAR10 | 1.3 | 0.9 | 1.1 | 54.8 | 58.6 | 27.5 |
| MNIST | 0.4 | 2.3 | 0.7 | 49.3 | 56.2 | 33.4 |
| SNLI | 13.0 | 22.0 | 17.6* | 34.7 | 62.5 | 28.9* |

Table 2: Calibration metrics ECE and MCE by non-DP (no clipping) and DP optimizers. *Note that the SNLI experiment uses the mix-up training as described in Section 5.5.

---

[7]An over-confident classifier, when predicting wrong at one data point, only reduces its accuracy a little but increases its loss significantly due to large $-\log(\pi_{y_i})$, since too little probability is assigned to the true class.

Throughout this paper, we use the GDP privacy accountant for the experiments, with `Private Vision` library (Bu et al., a) (improved on `Opacus`) and one P100 GPU. We cover a range of model architectures (including convolutional neural networks [CNN] and Transformers), batch sizes (from 32 to 1000), datasets (with sample size from 50,000 to 550,152), and tasks (including image and text classification). More details are available in Appendix C.

### 5.3 CIFAR10 image data with Vision Transformer

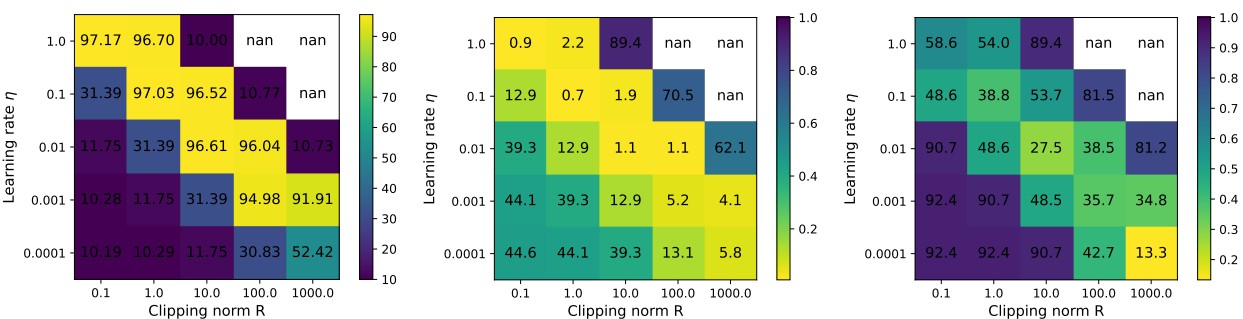

Figure 3: Ablation study on the accuracy, ECE and MCE (left to right) of CIFAR10 with ViT-base.

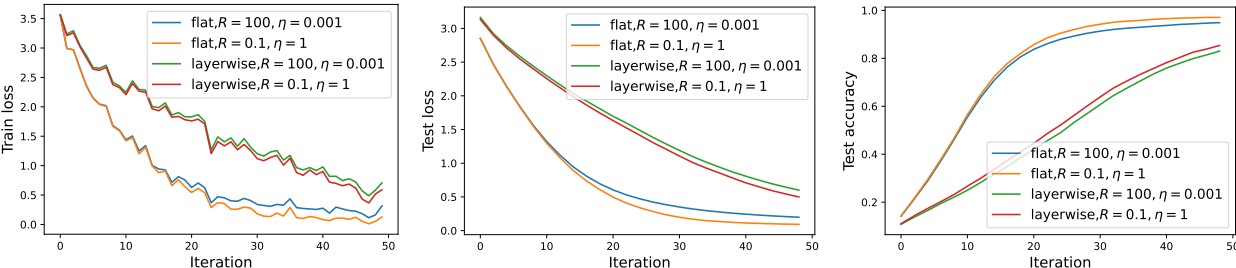

Figure 4: Performance on CIFAR10 with ViT-base, batch size 1000, noise scale 1.3, $(\epsilon, \delta) = (1.96, 10^{-5})$.

CIFAR10 is an image dataset, which contains 50000 training samples and 10000 test samples of $32 \times 32$ color images in 10 classes. We use the Vision Transformer (ViT-base, 86 million parameters) which is pre-trained on ImageNet and train with DP-SGD for a single epoch. This is one of state-of-the-art models for this DP task (Bu et al., a;b). From Figure 3[8], the best accuracy is achieved along the diagonal by small $R$ and large $\eta$, a phenomenon that is commonly observed in (Li et al., 2021; Bu et al., 2022). However, the calibration error (especially the MCE) is worse than the standard training in Table 2 and Figure 2. Additionally, the layerwise clipping can further slow down the optimization, as indicated by Theorem 1. We highlight that we choose $(R, \eta)$ proportioanlly, so that the total noise magnitude $\eta\sigma R$ is fixed for different hyperparameters.

On the other hand, DP training with larger $R$ can lead to significantly better calibration errors, while incurring a negligible reduction in the accuracy ($97.17 \to 96.61\%$). In Figure 5, the *reliability diagram* (DeGroot & Fienberg, 1983; Niculescu-Mizil & Caruana, 2005) displays the accuracy as a function of confidence. Graphically speaking, a calibrated classifier is expected to have blue bins close to the diagonal black dotted line. While the non-DP model is generally over-confident and thus not calibrated, the large $R$ clipping effectively achieves nearly perfect calibration, thanks to its Bayesian learning nature. In contrast, the classifier with small $R$ clipping is not only mis-calibrated, but also falls into 'bipolar disorder': it is either over-confident and inaccurate, or under-confident but highly accurate. This disorder is observed to different extent in all experiments in this paper.

---

[8]Note that the ablation study of $(\eta, R)$ is necessary and well-applied on DP optimization (see Figure 8 in (Li et al., 2021) and Figure 1 in (Bu et al., 2022)). Thus, besides the evaluation of accuracy, additionally evaluating the calibration error is almost free.

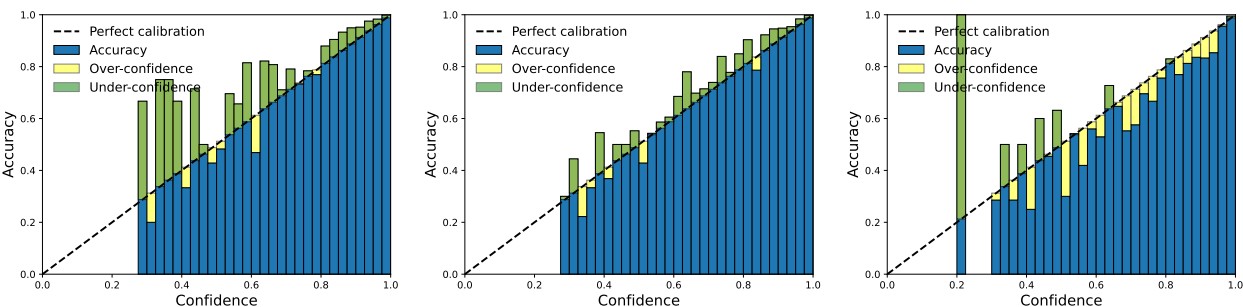

Figure 5: Reliability diagrams (left for non-DP; middle for DP with large $R$; right for DP with small $R$) on CIFAR10 with ViT-base.

### 5.4 MNIST image data with CNN model

On the MNIST dataset, which contains 60000 training samples and 10000 test samples of $28 \times 28$ grayscale images in 10 classes, we use the standard CNN in the DP libraries[9](Google; Facebook) (see Appendix C.1 for architecture) and train with DP-SGD but without pre-training. In Figure 6, DP training with both clipping norms is $(2.32, 10^{-5})$-DP, and has similar test accuracy (96% for small $R$ and 95% for large $R$), though the large $R$ leads to smaller loss (or NLL). In the right plot of Figure 6, we demonstrate how $R$ affects the accuracy and calibration, ceteris paribus, showing a clear accuracy-calibration trade-off based on 5 independent runs. Similar to Figure 5, large $R$ training again mitigates the mis-calibration in Figure 7.

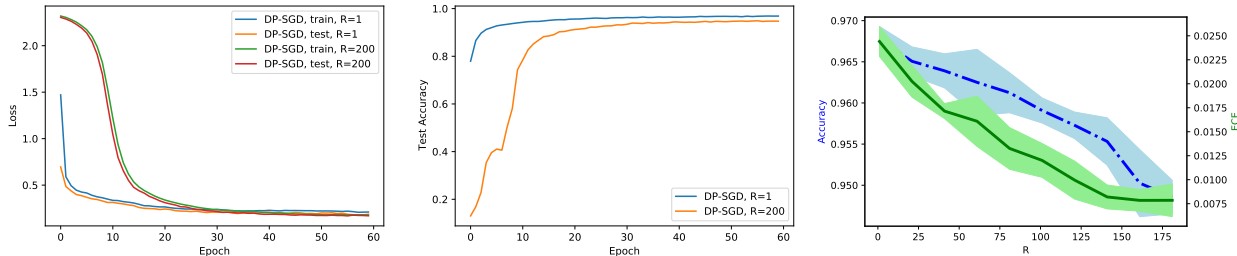

Figure 6: Loss (left), accuracy (middle), accuracy with ECE (right) on MNIST with 4-layer CNN under different clipping norms $R$, batch size 256, noise scale 1.1, learning rate $0.15/R$ for each $R$.

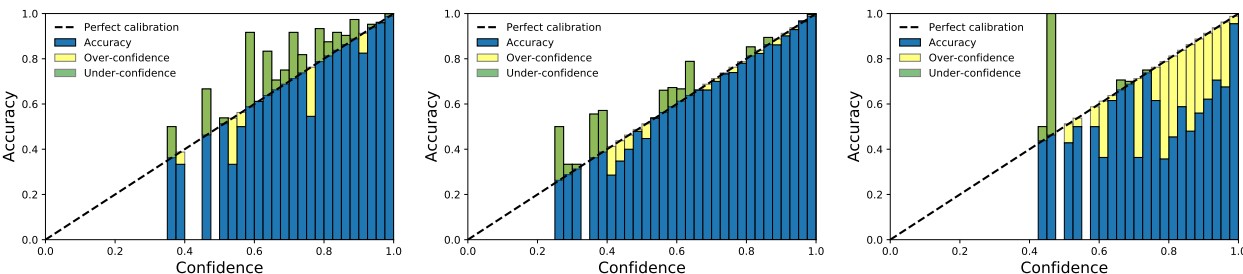

Figure 7: Reliability diagrams (left for non-DP; middle for large $R = 200$; right for small $R = 1$) on MNIST with 4-layer CNN.

---

[9]See https://github.com/tensorflow/privacy/tree/master/tutorials in Tensorflow and https://github.com/pytorch/opacus/blob/master/examples/mnist.py in Pytorch Opacus.

## 5.5 SNLI text data with BERT and mix-up training

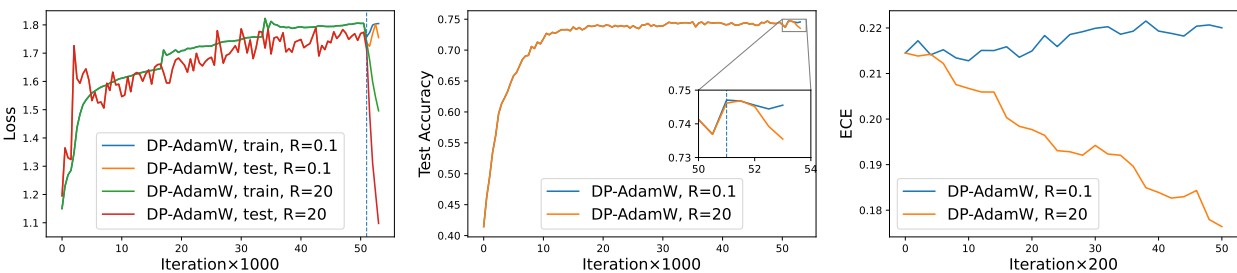

Figure 8: Loss (left), accuracy (middle) and calibration on SNLI with pre-trained BERT, batch size 32, learning rate 0.0005, noise scale 0.4, clipping norm are 0.1 or 20, $(\epsilon, \delta) = (1.25, 1/550152)$.

Stanford Natural Language Inference (SNLI) [10] is a collection of human-written English sentence paired with one of three classes: entailment, contradiction, or neutral. The dataset has 550152 training samples and 10000 test samples. We use the pre-trained BERT (Bidirectional Encoder Representations from Transformers) on `Opacus` tutorial[11], which gives a state-of-the-art privacy-accuracy result. Our BERT contains 108M parameters and we only train the last Transformer encoder, which has 7M parameters, using DP-AdamW. In particular, we use a **mix-up training**: we in fact train BERT with small $R$ for 3 epochs ($51.5 \times 10^3$ iterations, i.e. 95% of the training) and then use large $R$ for an additional 2500 iterations (the last 5% of the training). For comparison, we also train the same model with small $R$ for the entire training process of 54076 iterations.

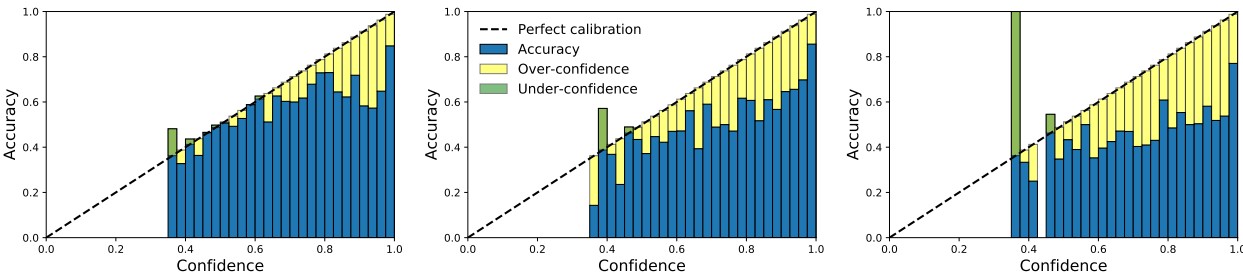

Figure 9: Reliability diagrams (left for non-DP; middle for large $R = 20$; right for small $R = 0.1$) on SNLI with BERT. Note that the large $R$ is only used for the last 2500 out of 54000 iterations.

Surprisingly, the existing DP optimizer does not minimize the loss at all, yet the accuracy still improves along the training. We again observe that large $R$ training has significantly better convergence than small $R$ (observe that when turned to large $R$ in the last 2500 steps, the test loss or NLL decreases significantly from 1.79 to 1.08, and the training loss or NLL decreases from 1.81 to 1.47; while keeping a small $R$ does not reduce the losses). The resulting models have similar accuracy: small $R$ has 74.1% accuracy; mix-up training has 73.1% accuracy; as baselines, non-DP has 85.4% accuracy and the entire training with large $R$ has 65.9% accuracy. All DP models have the same privacy ($\epsilon = 1.25, \delta = 1/550152$), and large $R$ training has much better calibration in Table 2. We remark that all hyperparameters are the same as in the `Opacus` tutorial.

## 5.6 Regression Tasks

On regression tasks, the performance measure and the loss function are unified as MSE. Figure 10 shows that DP training with large $R$ is comparable if not better than that with small $R$. We experiment on the California Housing data (20640 samples, 8 features) and Wine Quality (1599 samples, 11 features, run with full-batch

---

[10]We use SNLI 1.0 from `https://nlp.stanford.edu/projects/snli/`

[11]See `https://github.com/pytorch/opacus/blob/master/tutorials/building_text_classifier.ipynb`.

DP-GD). Especially, in the left plot of Figure 10, we observe that small $R$ training may incurs non-monotone convergence, as explained by Theorem 1, which is mitigated by the large $R$ training. Additional experimental details are available in Appendix C.4.

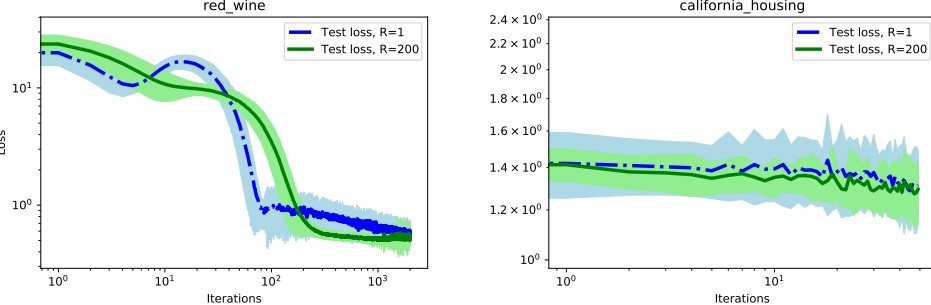

Figure 10: Performance of DP optimizers under different clipping norms on the Wine Quality and the California Housing datasets. Experimental details in Appendix C.4.

## 6 Discussion

In this paper, we provide a continuous-time convergence analysis for DP deep learning, via the NTK matrix, which applies to the general neural network architecture and loss function. We show that in such a regime, the noise addition only affects the privacy risk but not the convergence, whereas the per-sample clipping only affects the convergence and the calibration (especially with different choices of clipping thresholds), but not the privacy risk.

We then study the accuracy-calibration trade-off formed by the DP training with different clipping norms. We show that using a small clipping norm oftentimes trains the more accurate but mis-calibrated models, while a large clipping norm provides a comparably accurate yet much more calibrated model. In fact, several follow-up works have demonstrated that DP training with large $R$ is remarkably accurate and well-calibrated on large transformers with $> 10^8$ parameters (Zhang et al., 2022), and it significantly mitigates the unfairness on various tasks (Esipova et al., 2022), while preserving privacy.

A future direction is to study the discrete time convergence when both the learning rate and added noise are not small. One immediate observation is that the noise addition will have an effect on the convergence in this case, which needs further investigation. In addition, the analysis of commonly-used *mini-batch optimizers* is also interesting, since for those optimizers, the training dynamics is no longer deterministic and instead stochastic differential equation will be used for analsis. Lastly, the inconsistency between the cross-entropy loss and the prediction accuracy, as well as the connection to the calibration issue are intriguing; their theoretical understanding awaits future research.

## Acknowledgement

We would like to thank Weijie J. Su, Janardhan Kulkarni, Om Thakkar, and Gautam Kamath for constructive and stimulating discussions around the global clipping function. We also thank the `Opacus` team for maintaining this amazing library. This work was supported in part by NIH through R01GM124111 and RF1AG063481.

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

## A  Linear Algebra Facts

**Fact A.1.** The product $A = M_1 M_2$, where $M_1$ is a symmetric and positive matrix and $M_2$ a positive diagonal matrix, is positive definite in eigenvalues but is non-symmetric in general (unless the diagonal matrix is constant) and non-positive in quadratic forms.

*Proof of Fact A.1.* To see the non-symmetry of $A$, suppose there exists $i, j$ such that $(M_2)_{jj} \neq (M_2)_{ii}$, then

$$(M_1 M_2)_{ij} = \sum_k (M_1)_{ik}(M_2)_{kj} = (M_1)_{ij}(M_2)_{jj} = (M_1)_{ji}(M_2)_{jj},$$

$$(M_1 M_2)_{ji} = (M_1)_{ji}(M_2)_{ii} \neq (M_1)_{ji}(M_2)_{jj}.$$

Hence $A$ is not symmetric and positive definite. To see that $A$ may be non-positive in the quadratic form, we give a counter-example.

$$M_1 = \begin{pmatrix} 1 & 1 \\ 1 & 2 \end{pmatrix}, M_2 = \begin{pmatrix} 1 & 0 \\ 0 & 0.1 \end{pmatrix}, A = M_1 M_2 = \begin{pmatrix} 1 & 0.1 \\ 1 & 0.2 \end{pmatrix}, (1, -2)A \begin{pmatrix} 1 \\ -2 \end{pmatrix} = -0.4.$$

To see that $A$ is positive in eigenvalues, we claim that an invertible square root $M_1^{1/2}$ exists as $M_1$ is symmetric and positive definite. Now $A$ is similar to $(M_1^{1/2})^{-1}AM_1^{1/2} = M_1^{1/2}M_2 M_1^{1/2}$, hence the non-symmetric $A$ has the same eigenvalues as the symmetric and positive definite $M_1^{1/2}M_2 M_1^{1/2}$. □

**Fact A.2.** Matrix with all eigenvalues positive may be non-positive in quadratic form.

*Proof of Fact A.2.*

$$A = \begin{pmatrix} -1 & 3 \\ -3 & 8 \end{pmatrix}, (1, 0)A \begin{pmatrix} 1 \\ 0 \end{pmatrix} = -1,$$

though eigenvalues of $A$ are $\frac{1}{2}(7 \pm 3\sqrt{5}) > 0$. □

**Fact A.3.** Matrix with positive quadratic forms may have non-positive eigenvalues.

*Proof of Fact A.3.*

$$A = \begin{pmatrix} 1 & 1 \\ -1 & 1 \end{pmatrix}, (x, y)A \begin{pmatrix} x \\ y \end{pmatrix} = x^2 + y^2 > 0,$$

but eigenvalues of $A$ are $1 \pm i$, not positive nor real. Actually, all eigenvalues of $A$ always have positive real part. □

**Fact A.4.** Sum of products of positive definite (symmetric) matrix and positive diagonal matrix may have zero or negative eigenvalues.

*Proof of Fact A.4.*

$$\mathbf{H}_1 = \begin{pmatrix} 8/9 & 2 \\ 2 & 7 \end{pmatrix}, \quad \mathbf{C}_1 = \begin{pmatrix} 0.9 & 0 \\ 0 & 0.4 \end{pmatrix}, \quad \mathbf{H}_2 = \begin{pmatrix} 3 & 2 \\ 2 & 2 \end{pmatrix}, \quad \mathbf{C}_2 = \begin{pmatrix} 0.1 & 0 \\ 0 & 0.6 \end{pmatrix}.$$

Although $\mathbf{H}_j$ are positive definite, $\mathbf{H}_1\mathbf{C}_1 + \mathbf{H}_2\mathbf{C}_2$ has a zero eigenvalue. Further, if $\mathbf{H}_1[1, 1] = 0.7$, $\mathbf{H}_1\mathbf{C}_1 + \mathbf{H}_2\mathbf{C}_2$ has a negative eigenvalue. □

# B  Details of Main Results

## B.1  Proofs of main results

*Proof of Fact 4.1.* Expanding the discrete dynamic in equation 4.1 as $\mathbf{w}(k+1) = \mathbf{w}(k) - \frac{\eta}{n}\sum_i \nabla_{\mathbf{w}}\ell_i C_i - \frac{\eta\sigma R}{n}\mathcal{N}(0,1)$, and chaining it for $r \geq 1$ times, we obtain

$$\mathbf{w}(k+r) - \mathbf{w}(k) = -\sum_{j=0}^{r-1}\frac{\eta}{n}\sum_i \nabla_{\mathbf{w}}\ell_i(\mathbf{w}(k+j))C_i - \sum_{j=0}^{r-1}\frac{\eta\sigma R}{n}\mathcal{N}(0,1).$$

In the limit of $\eta \to 0$, we re-index the weights $\mathbf{w}$ by time, with $t = k\eta$ and $s = r\eta$. Then consider the above equation at time $t+s$ and $t$: the left hand side becomes $\mathbf{w}(t+s) - \mathbf{w}(t)$; the first summation on the right hand side converges to $-\frac{1}{n}\int_t^{t+s}\sum_i \nabla_{\mathbf{w}}\ell_i(\tau)C_i(\tau)d\tau$, as long as the integral exists. This can be seen as a numerical integration with the rectangle rule, using $\eta$ as the width, $j$ as the index, and $\frac{1}{n}\int_t^{t+s}\sum_i \nabla_{\mathbf{w}}\ell_i(\tau)C_i(\tau)$; similarly, the second summation $J(\eta) = \sum_{j=0}^{r-1}\frac{\eta\sigma R}{n}\mathcal{N}(0,1)$ has

$$\mathbb{E}[J(\eta)] = 0 \quad \text{and} \quad \mathrm{Var}(J(\eta)) = \frac{\sigma^2 R^2 \eta^2}{n^2}r = \eta s \frac{\sigma^2 R^2}{n^2} \to 0, \text{ as } \eta \to 0.$$

Therefore, as $\eta \to 0$, the discrete stochastic dynamic equation 4.1 becomes the integral

$$\mathbf{w}(t+s) - \mathbf{w}(t) = -\frac{1}{n}\int_t^{t+s}\sum_i \nabla_{\mathbf{w}}\ell_i(\tau)C_i(\tau)d\tau.$$

This integral converges to a deterministic gradient flow, as $s \to 0$, given by

$$\dot{\mathbf{w}}(t) \equiv \lim_{s\to 0}\frac{\mathbf{w}(t+s) - \mathbf{w}(t)}{s} = -\frac{1}{n}\lim_{s\to 0}\frac{\int_t^{t+s}\sum_i \nabla_{\mathbf{w}}\ell_i(\tau)C_i(\tau)d\tau}{s}.$$

which corresponds to the ordinary differential equations equation 4.2. $\qquad\square$

*Proof of Theorem 1.* We prove the statements using the derived gradient flow dynamics equation 4.2.

For Statement 1, from our narrative in Section 4.2, we know that the flat clipping algorithm has $\mathbf{H}(t)\mathbf{C}(t)$ as its NTK. Since $\mathbf{H}(t)$ is positive definite and $\mathbf{C}(t)$ is a positive diagonal matrix, by Fact A.1, the product $\mathbf{H}(t)\mathbf{C}(t)$ is positive in eigenvalues, yet may be asymmetric and not positive in quadratic form in general.

Similarly, for Statement 2, we know the NTK of layerwise clipping has the form $\sum_r \mathbf{H}_r(t)\mathbf{C}_r(t)$, which by Fact A.4 is asymmetric in general, and may be not positive in quadratic form nor positive in eigenvalues.

For Statement 3, by the training dynamics equation 4.3 for the flat clipping algorithm and equation 4.4 for the layerwise clipping, we see that $\dot{L}$ equal the negation of a quadratic form of the corresponding NTK. By statement 1 & 2 of this theorem, such quadratic form may not be positive at all $t$, and hence the loss $L(t)$ is not guaranteed to decrease monotonically.

Lastly, for Statement 4, suppose $L(t)$ converges in the sense that $\dot{L} = 0 = \frac{\partial L}{\partial \boldsymbol{f}}\dot{\boldsymbol{f}}$. Suppose we have $L > 0$, then $\frac{\partial L}{\partial \boldsymbol{f}} \neq 0$ since $L$ is convex in the prediction $\boldsymbol{f}$. In this case, we know $\dot{\boldsymbol{f}} = 0$. Observe that

$$0 = \dot{\boldsymbol{f}} = \frac{\partial \boldsymbol{f}}{\partial \mathbf{w}}\frac{\partial \mathbf{w}}{\partial t} = -\frac{\partial \boldsymbol{f}}{\partial \mathbf{w}}\frac{\partial \boldsymbol{f}}{\partial \mathbf{w}}^\top \frac{\partial L}{\partial \boldsymbol{f}}^\top .$$

For the flat clipping, the NTK matrix, $\frac{\partial \boldsymbol{f}}{\partial \mathbf{w}}\frac{\partial \boldsymbol{f}}{\partial \mathbf{w}}^\top = \mathbf{H}\mathbf{C}$ is positive in eigenvalues (by Statement 1), so it could only be the case that $\frac{\partial L}{\partial \boldsymbol{f}} = \mathbf{0}$, contradicting to our premise that $L > 0$. Therefore we know $L = 0$ as long as it converges for the flat clipping. On the other hand, for the layerwise clipping, the NTK may be not positive in eigenvalues. Hence it is possible that $L \neq 0$ when $\dot{L} = 0$.

$\qquad\square$

*Proof of Theorem 2.* The proof is similar to the previous proof and thus omitted.

$\qquad\square$

# C Experimental Details

## C.1 MNIST

For MNIST, we use the standard CNN in `Tensorflow Privacy` and `Opacus`, as listed below. The training hyperparameters (e.g. batch size) in Section 5.4 are exactly the same as reported in `https://github.com/tensorflow/privacy/tree/master/tutorials`, which gives 96.6% accuracy for the small $R$ clipping in Tensorflow and similar accuracy in Pytorch, where our experiments are conducted. The non-DP network is about 99% accurate. Notice the tutorial uses a different privacy accountant than the GDP that we used.

```
class SampleConvNet(nn.Module):
    def __init__(self):
        super().__init__()
        self.conv1 = nn.Conv2d(1, 16, 8, 2, padding=3)
        self.conv2 = nn.Conv2d(16, 32, 4, 2)
        self.fc1 = nn.Linear(32 * 4 * 4, 32)
        self.fc2 = nn.Linear(32, 10)

    def forward(self, x):
        # x of shape [B, 1, 28, 28]
        x = F.relu(self.conv1(x))  # -> [B, 16, 14, 14]
        x = F.max_pool2d(x, 2, 1)  # -> [B, 16, 13, 13]
        x = F.relu(self.conv2(x))  # -> [B, 32, 5, 5]
        x = F.max_pool2d(x, 2, 1)  # -> [B, 32, 4, 4]
        x = x.view(-1, 32 * 4 * 4)  # -> [B, 512]
        x = F.relu(self.fc1(x))  # -> [B, 32]
        x = self.fc2(x)  # -> [B, 10]
        return x
```

## C.2 CIFAR10 with Vision Transformer

In Section 5.3, we adopt the model from TIMM library. In addition to Figure 2 and Figure 5, we plot in Figure 11 the distribution of prediction probability on the true class, say $[\boldsymbol{\pi}_i]_{y_i}$ for the $i$-th sample (notice that Figure 2 plots $\max_k[\boldsymbol{\pi}_i]_k$). Clearly the small $R$ clipping gives overly confident prediction: almost half of the time the true class is assigned close to zero prediction probability. The large $R$ clipping has a more balanced prediction probability that is less concentrated to 1.

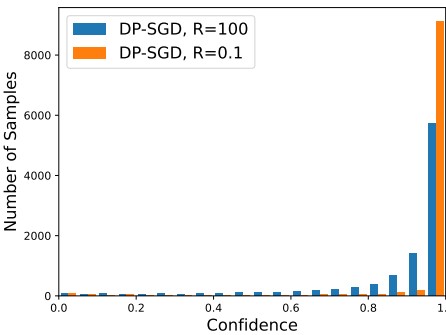

Figure 11: Prediction probability on the true class on CIFAR10 with Vision Transformer.

### C.3 SNLI with BERT model

In Section 5.5, we use the model from `Opacus` tutorial in `https://github.com/pytorch/opacus/blob/master/tutorials/building_text_classifier.ipynb`. The BERT architecture can be found in `https://github.com/pytorch/opacus/blob/master/tutorials/img/BERT.png`.

To train the BERT model, we do the standard pre-processing on the corpus (tokenize the input, cut or pad each sequence to MAX_LENGTH = 128, and convert tokens into unique IDs). We train the BERT model for 3 epochs. Similar to Appendix C.2, in addition to Figure 8 and Figure 9, we plot the distribution of prediction probability on the true class in Figure 12. Again, the small $R$ clipping is overly confident, with probability masses concentrating on the two extremes, yet the large $R$ clipping is more balanced in assigning the prediction probability.

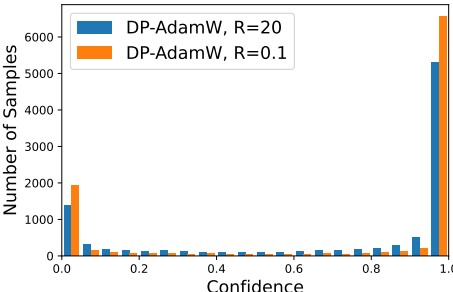

Figure 12: Histogram of predicted confidence on the true class on SNLI with BERT using large and small clipping norms.

### C.4 Regression Experiments

We experiment on the Wine Quality[12] (1279 training samples, 320 test samples, 11 features) and California Housing[13] (18576 training samples, 2064 test samples, 8 features) datasets in Section 5.2. For the California Housing, we use DP-Adam with batch size 256. Since other datasets are not large, we use the full-batch DP-GD.

Across all the two experiments, we set $\delta = \frac{1}{1.1 \times \text{training sample size}}$ and use the four-layer neural network with the following structure, where `input_width` is the input dimension for each dataset:

```python
class Net(nn.Module):
    def __init__(self, input_width):
        super(StandardNet, self).__init__()
        self.fc1 = nn.Linear(input_width, 64, bias = True)
        self.fc2 = nn.Linear(64, 64, bias = True)
        self.fc3 = nn.Linear(64, 32, bias = True)
        self.fc4 = nn.Linear(32, 1, bias = True)

    def forward(self, x):
        x = F.relu(self.fc1(x))
        x = F.relu(self.fc2(x))
        x = F.relu(self.fc3(x))
        return self.fc4(x)
```

---

[12]`http://archive.672ics.uci.edu/ml/datasets/Wine+Quality`
[13]`http://lib.stat.cmu.edu/datasets/houses.zip`

The California Housing dataset is used to predict the mean price value of owner-occupied home in California. We train with DP-Adam, noise $\sigma = 1$, clipping norm 1, and learning rate 0.0002. We also trained a non-DP GD with the same learning rate. The GDP accountant gives $\epsilon = 4.41$ after 50 epochs / 3650 iterations.

The UCI Wine Quality (red wine) dataset is used to predict the wine quality (an integer score between 0 and 10). We train with DP-GD, noise $\sigma = 35$, clipping norm 2, and learning rate 0.03. We also trained a non-DP GD with learning rate 0.001. The GDP accountant gives $\epsilon = 4.40$ after 2000 iterations.

The California Housing and Wine Quality experiments are conducted in 30 independent runs. In Figure 10, the lines are the average losses and the shaded regions are the standard deviations.

## D  Global clipping and code implementation

In an earlier version of this paper, we proposed a new per-sample gradient clipping, termed as the *global clipping*. The global clipping computes $C_{global,i} = \mathbb{I}(\|\boldsymbol{g}^{(i)}\| \leq R)$, i.e. only assigning 0 or 1 as the clipping factors to each per-sample gradient.

As demonstrated in equation 2.2, our global clipping works with any DP optimizers (e.g., DP-Adam, DP-RMSprop, DP-FTRL(Kairouz et al., 2021), DP-SGD-JL(Bu et al., 2021a), etc.), with *identical computational complexity* as the existing per-sample clipping $C_i = \min(R/\|\boldsymbol{g}^{(i)}\|, 1)$. Building on top of the Pytorch `Opacus`[14] library, we only need to add one line of code into

https://github.com/pytorch/opacus/blob/master/opacus/per_sample_gradient_clip.py

To understand our implementation, we can equivalently view

$$C_{global,i} = \begin{cases} 1 & \text{if } C_i = 1 \iff \|\boldsymbol{g}^{(i)}\| < R \iff \min(R/\|\boldsymbol{g}^{(i)}\|, 1) = 1 \\ 0 & \text{if } C_i = R/\|\boldsymbol{g}^{(i)}\| \iff \|\boldsymbol{g}^{(i)}\| \geq R \iff \min(R/\|\boldsymbol{g}^{(i)}\|, 1) = R/\|\boldsymbol{g}^{(i)}\| \end{cases}$$

In this formulation, we can easily implement our global clipping by leveraging the `Opacus==0.15` library (which already computes $C_i$). This can be realized in multiple ways. For example, we can add the following one line after line 179 (within the for loop),

```
clip_factor=(clip_factor>=1).float()
```

At high level, global clipping does not clip small per-sample gradients (in terms of magnitude) and completely remove large ones. This may be beneficial to the optimization, since large per-sample gradients often correspond to samples that are hard-to-learn, noisy or adversarial. It is important to set a large clipping norm $R$ for the global clipping, so that the information from small per-sample gradients are not wasted. However, using a large clipping norm makes the global clipping similar to the existing clipping, basically not clipping most of the per-sample gradients. We confirm that empirically, with large clipping norm, applying the global clipping and existing clipping have negligible difference on the convergence and calibration.

---

[14]see https://github.com/pytorch/opacus as for 2021/09/09.

