# OpenReview forum: "On the Convergence and Calibration of Deep Learning with Differential Privacy"
_TMLR — Accepted by TMLR_

### Review · Reviewer_VPen · 2023-03-30

**Summary Of Contributions:**

The objective of this paper is to better understand the influence of the DP training parameters. The authors clearly claim that:
- Gaussian noise has no significant effect on convergence/calibration.
- Small clipping improves training accuracy but impacts calibration.
- Large clipping improves calibration.

These statements are supported by a theoretical analysis utilizing neural tangent kernel matrices, which is further verified through numerical examples.

**Audience:**

Yes

**Broader Impact Concerns:**

None.

**Claims And Evidence:**

Yes

**Requested Changes:**

Clarifications:

* I am uncertain about the comment in Section~5, which states that the noise is of small magnitude in the described setting. Since the updates involve two terms: (1) the gradient near the loss function's optimum, whose norm approaches $0$, versus (2) the Gaussian noise of norm $(\mathrm{dim}\times \eta\sigma R / n)$, it is not immediately clear to me that the noise has a minor effect on the accuracy -- especially because modern machine learning models may have millions of parameters, this scaling with the dimension can lead to significantly less accurate models.

* In Section~4.5, I am not convinced by the comparison with SGLD using a different factor $\eta$. As the authors mentioned, the noise is much larger when $\eta\to 0$. The dynamics are not the same, and the Stochastic Gradient Langevin Diffusion arises from the Langevin diffusion discretization. The objective of SGLD is to sample parameters $w\propto \exp(-L)$ to estimate the predictive distribution and quantify the uncertainty. I am not entirely convinced by the analogy.

* Regarding the proof of Statement~4 (page 16), the connection between the convergence of $L$ and the derivative $L'=0$ would require more explanations. For instance, if (1) $L(t)=log(t)$, or (2) $L'(t)=\sum_{n=0}^{\infty}\varphi(2^n (t-n)) 1_{[n,n+1)}(t)$ with $\varphi(x)=[2 relu(x) - 4 relu(x-1/2)]_{+}$, then L is continuously differentiable and lim L = 1, but L' does not tends to zero.


Formulation:

* Please, could you provide the assumptions on $f$ for better clarity.

* It would be helpful to use a consistent notation for the function $C_i$ throughout the paper. Would it be possible to use a single notation for the function, and define it clearly at the beginning of the paper? (Instead of: $C_i$, $C_i(t)$, $C_i(R)$, $C_i(w)$).



**Strengths And Weaknesses:**

STRENGTHS.

+ The paper is written in a concise style, with clear statements that are easily identifiable in the abstract.

+ This paper conducts a thorough investigation to improve our comprehension of the impact of DP (Differential Privacy).


WEAKNESSES.

My main concern about this paper is the lack of rigor in certain parts. For instance, I couldn't find any differentiability assumptions on $f$, yet there are results based on its differential. I believe that the paper would benefit greatly if a complete set of assumptions were established
In the same vein, some proofs are a bit hasty, such as the proof of Fact 4.1 or (Theorem 1->Statement 4). Please, could the authors provide more details?

---

> ### Author Response · Authors · 2023-04-25
>
> We thank the reviewer for the comment! We have modified the main text to enhance the rigor and clarity of function C_i. Below is a point-to-point response.
>
> Weakness 1: For instance, I couldn't find any differentiability assumptions on f, yet there are results based on its differential.
>
> **Response** We agree that the differentiability of f is assumed, as is the case in most literature of deep learning convergence: the neural network f is always assumed to be differentiable so as to leverage the back-propagation algorithm (a chain rule of derivatives). In practice, the differentialbility is algorithmically taken care of by auto-differentiation systems such as Tensorflow or Pyrotch. For example, Eqn (1) and Sec 4 of [1], Fact 2.6 of [2], Eqn (1) of [3] (differentiability of loss L implies differentiability of model f). However, this indeed lacks rigor in the sense some components of the model may not be differentiable, such as the ReLU activation function (which only has sub-gradient not gradient at 0). We have added Footnote 2 for this discussion.
>
> Weakness 2: some proofs are a bit hasty, such as the proof of Fact 4.1 or (Theorem 1->Statement 4).
>
> **Response** We have polished the proof of Fact 4.1 when we move between the continuous-time and discrete-time. We are happy to provide more details if the reviewer would like to be more specific about which parts is 'hasty'.
>
> Clarifications 1: I am uncertain about the comment in Section~5, which states that the noise is of small magnitude in the described setting.
>
> **Response** We thank the reviewer for the insightful comment! As we discussed in Sec 5 and in Fig 1, the per-parameter noise (ignoring the model dimension) is $\eta\sigma R/n$, which is very small and insignificant *in practice*. We agree that since the model dimension can be of millions of parameters, it seems that the total noise magnitude is large. However, this is not the case in practice when larger model usually gives better accuracy (e.g. [4] going from small vision transformer to 300M BEiT gives better accuracy; [5] going from GPT2-small (124M) to GPT2-large (774M) gives better BLEU score). One promising explanation is that the effective model dimension is much smaller than the model dimension. We quote from [6] that "gradients obtained during fine-tuning are
> mostly controlled by a few principal components". We have added this discussion to Sec 5, Footnote 6.
>
> Clarifications 2: In Section~4.5, I am not convinced by the comparison with SGLD using a different factor
>
> **Response** We agree that the dynamics of un-clipped DP-SGD and SGLD are different, and that they are used for different purposes. However, we assure the reviewer that the un-clipped DP-SGD is equivalent to SGLD **in the discrete time**. Here DP-SGD (when clipping is not effective, i.e. $C_i\approx 1 \forall i$) is
> $$w(k+1)-w(k)=-\frac{\eta_{DPSGD}}{n}\left(\sum_i\frac{\partial l_i}{\partial w}+\sigma R N(0,I)\right)$$
> SGLD (https://en.wikipedia.org/wiki/Stochastic_gradient_Langevin_dynamics#Formal_definition) is
> $$w(k+1)-w(k)=-\frac{\eta_{SGLD}N}{2n}\left(\sum_i\frac{\partial l_i}{\partial w}\right)+\sqrt{\eta_{SGLD}}N(0,I)$$
> if we ignore the prior term log p. Clearly, DP-SGD (with the right combination of hyperparameters) is a special form of SGLD by setting $\eta_{DPSGD}=\eta_{SGLD}N/2$ and $\sigma R\frac{N}{2n}=1/\sqrt{\eta_{SGLD}}$. We will add this formally to Sec 4.5 if the reviewer agrees.
>
> Clarification 3: Regarding the proof of Statement~4 (page 16), the connection between the convergence of L and the derivative L'=0 would require more explanations.
>
> **Response** We agree that one needs, e.g. uniform convergence, to establish that L converges means L'=0. We have modified the State 4 of Theorem 1 and added Footnote 5 to explain.
>
>
> [1] Du, Simon, Jason Lee, Haochuan Li, Liwei Wang, and Xiyu Zhai. "Gradient descent finds global minima of deep neural networks." In International conference on machine learning, pp. 1675-1685. PMLR, 2019.
>
> [2] Allen-Zhu, Zeyuan, Yuanzhi Li, and Zhao Song. "A convergence theory for deep learning via over-parameterization." In International Conference on Machine Learning, pp. 242-252. PMLR, 2019.
>
> [3] Xie, Zeke, Issei Sato, and Masashi Sugiyama. "A diffusion theory for deep learning dynamics: Stochastic gradient descent exponentially favors flat minima." arXiv preprint arXiv:2002.03495 (2020).
>
> [4] Bu, Zhiqi, Jialin Mao, and Shiyun Xu. "Scalable and Efficient Training of Large Convolutional Neural Networks with Differential Privacy." In Advances in Neural Information Processing Systems.
>
> [5] Li, Xuechen, Florian Tramer, Percy Liang, and Tatsunori Hashimoto. "Large Language Models Can Be Strong Differentially Private Learners." In International Conference on Learning Representations.
>
> [6] Li et al. "When Does Differentially Private Learning Not Suffer in High Dimensions?." Advances in Neural Information Processing Systems 35 (2022): 28616-28630.

---

### Review · Reviewer_tCgp · 2023-03-31

**Summary Of Contributions:**


The paper aims at analyzing the effect of clipping and noising the gradient on the convergence of DP-SGD. In particular, the authors study per-layer and flat clipping techniques and their effect on the loss value, the accuracy, and the calibration of the model.

The authors find that noise addition does not change the dynamics of the gradient flow, while clipping does when the clipping norm is small. To explain this fact, the authors draw a connection with the neural tangent kernel (NTK) and argue that clipping can change its positiveness conditions. (Theorem 1).

The authors present experiments on CIFAR, MNIST, SNLI, and regression tasks showing the effects of the different clipping techniques and the noise on calibration, loss, and accuracy.



**Audience:**

Yes

**Broader Impact Concerns:**

I don't see any major concerns, the paper is trying to advance the understanding of private algorithms for learning models from sensitive data.

**Claims And Evidence:**

No

**Requested Changes:**

Addressing the following questions/concerns could strengthen the work.

1. If I understand correctly, from the theoretical statements it is always better to use a large enough clipping norm so that very few gradients (or none) are clipped. Throughout the paper the authors advocate for using "large R" (clipping norm). However, figure 3 shows that large R can also have a bad MCE and ECE. Thus “large R” is not a very helpful way of characterizing or selecting this parameter. Can the authors comment on how one could tune this value or what does "large" mean for a specific problem instance?

2. Is there any information that can be derived from the NTK analysis of the gradient flow that could help design the clipping strategy and value?

3. I am confused about Figure 1, why would different randomized descent trajectories, (furthermore with different noise scales) have the exact same loss values?

_Minor changes_: n and p in Lemma 2.2. Are not defined


**Strengths And Weaknesses:**

**Strengths**

- This paper proposes to study a fundamental problem in the literature: understanding the dynamics of DP-SGD is an important open problem given the proliferation of machine learning models trained on sensitive data. It does so by drawing connections to the NTK, which has not been widely explored to the best of my knowledge.

- The experiments on calibration are novel and have an actionable take for machine learning practitioners when deciding the trade-off between clipping/noise value.

**Weaknesses**

- It is interesting to see the connection between convergence and clipping through the positiveness of the NTK, however the paper does not provide any guidance on how to use this information, but rather confirms two known facts, in the literature: (1) gradient descent with noisy but unbiased gradients still converges, and (2) that over clipping can destroy the convergence. [1].

- I think studying calibration is very interesting but I could not understand from the paper the connection with the NTK.
- Theorem 2 I believe just follows from Fact 4.1.

[1] Song, Shuang, Om Thakkar, and Abhradeep Thakurta. "Characterizing private clipped gradient descent on convex generalized linear problems." arXiv preprint arXiv:2006.06783 (2020).

---

> ### Author Response · Authors · 2023-04-25
>
> We thank the reviewer for the comment! We have modified the main text, e.g. Lemma 2.2, to enhance the clarity. Below is a point-to-point response.
>
> Weakness 1: however the paper does not provide any guidance on how to use this information, but rather confirms two known facts, in the literature: (1) gradient descent with noisy but unbiased gradients still converges, and (2) that over clipping can destroy the convergence.
>
> **Response** Our contribution is to initiate a new observation between DP and NTK. With respect to "how to use this information", we propose to use a larger R (compared to the R that achieves the best accuracy) to improve the NTK positivity and thus the calibration (see Table 2, MCE column). This usage is detailed in Sec 4.3 & 4.4.
>
> We highlight three key differences between our findings and the known facts brought up by the reviewer. (1)  Song et al. only work on convex loss (i.e. only linear model) whereas our training dynamics works for any loss, including non-convex (w.r.t parameters) one that corresponds to deep learning; (2) the observation on calibration is new and only related to NTK in our work; (3) we derive the separation of noise and clipping in DP dynamics in Fact 4.1.
>
> Weakness 2: I think studying calibration is very interesting but I could not understand from the paper the connection with the NTK.
>
> **Response** We appreciate the reviewer’s acknowledgement in the importance of studying calibration. Our observation and analysis have suggested that (1) calibration of DP optimization is related to NTK and (2) calibration is *only* related to NTK, i.e. the per-sample gradient clipping, not the noising, as we show in Fact 4.1 and empirically in Figure 1.
>
> To see the first point, we argue that calibration is a property affected by the convergence, which is determined by the DP dynamics, or specifically, the NTK. It can be shown that with a sufficiently large R, the clipping is not effective, and the DP-SGD reduces to a noisy gradient descent in the discrete time, one that is very similar to SGLD under the right hyperparameters, as we discussed in Sec 4.5. Therefore, it is possible (and empirically supported) that DP-SGD with larger R has better calibration as SGLD is known to effectively train Bayesian neural networks, which are well-calibrated as it accounts for the uncertainty in prediction. We will surely add this discussion to the next revision if the reviewer agrees.
>
> Weakness 3: Theorem 2 I believe just follows from Fact 4.1.
>
> **Response** We agree that Theorem 2 follows from Fact 4.1, though these two results serve different purposes: Fact 4.1  formally separates the noising and the clipping in the continuous time; Theorem 2 interprets the consequence of such separation as it gives explicit description of training/optimization behaviors whereas Fact 4.1 does not.
>
> Requested Change 1: Throughout the paper the authors advocate for using "large R" (clipping norm). However, figure 3 shows that large R can also have a bad MCE and ECE. Thus “large R” is not a very helpful way of characterizing or selecting this parameter. Can the authors comment on how one could tune this value or what does "large" mean for a specific problem instance?
>
> **Response** We agree that a *moderately large* R is beneficial for calibration as we see from Figure 3, but one cannot expect an arbitrarily large R will be beneficial. In practice, one need to conduct an ablation study of learning rate $\eta$ and clipping threshold R, like in Figure 8 of [1] and Figure 1 of [2]. While in previous literature, the ablation study is to find the optimal hyperparameters for the accuracy, this ablation study can be used for free to evaluate the calibration. We thank the reviewer for this practical comment and will add this guideline to the main text.
>
> [1] Li, Xuechen, Florian Tramer, Percy Liang, and Tatsunori Hashimoto. "Large Language Models Can Be Strong Differentially Private Learners." In International Conference on Learning Representations.
>
> [2] Bu, Zhiqi, Yu-Xiang Wang, Sheng Zha, and George Karypis. "Automatic clipping: Differentially private deep learning made easier and stronger." arXiv preprint arXiv:2206.07136 (2022).

---

> ### Author Response · Authors · 2023-04-25
>
> Requested Change 2: Is there any information that can be derived from the NTK analysis of the gradient flow that could help design the clipping strategy and value?
>
> **Response** Our NTK analysis mostly focuses on explaining and understanding the behavior of the convergence, which may further influence the potential trade-off between various metrics of performance under different values and strategies of clipping. Therefore, it may not be directly applicable to designing a “best option” for a specific application. This is in fact the nature of this kind of theoretical analysis - we enumerate possibilities of conditions, and demonstrate the dynamics of training under each of those conditions. However, the choice of clipping strategy and value may really depend on the actual condition for a specific application scenario, and our theory indeed makes the practitioner aware of the tradeoff, which will help in a qualitative way for them to make better informed decisions.
>
>
> Requested Change 3: I am confused about Figure 1, why would different randomized descent trajectories, (furthermore with different noise scales) have the exact same loss values?
>
> **Response** We apologize for the potential confusion here. Just to clarify that the losses are not the same, though they seem to be close to each other in the current scale in Figure 1. In the final version, we would like to add a zoom-in plot with scale so that one can distinguish the difference among different trajectories in the appendix, together with the code that can reproduce the same plot. To give some more insight, those curves are close to each other because we force the same random seed at the beginning of each iteration among different runs. Specifically, we draw the same standard Gaussian noise, then scale it with different values $\sigma$ as the final noise added. This is to eliminate the potential difference from uncontrolled random realizations, in the hope for reproducibility and a fair comparison.

---

### Review · Reviewer_8Xk4 · 2023-04-11

**Summary Of Contributions:**

This paper analyzes the convergence of differentially private (DP) training through the lens of neural tangent kernel (NTK), for arbitrary network architectures and loss functions. The paper shows that per-sample gradient clipping affects the convergence and calibration, while noise addition only affects the privacy risk.


**Audience:**

Yes

**Claims And Evidence:**

Yes

**Requested Changes:**

1. Please provide empirical validation of its theoretical results on real-world datasets, and it remains to be seen whether the observed trade-off between accuracy and calibration holds in practice.
2. Add more experiments, the data and the NN structure in the paper are quite small. More experiments and large data (such as ImageNet) are needed
3. Compare with SOTA methods  such as large batch and add more baselines.

**Strengths And Weaknesses:**

The contributions of this paper are:

- The paper formulates a continuous time analysis of differentially private (DP) training through the lens of neural tangent kernel (NTK), which characterizes the per-sample gradient clipping and the noise addition in DP training, for arbitrary network architectures and loss functions.
- The paper shows that the noise addition only affects the privacy risk but not the convergence or calibration, whereas the per-sample gradient clipping (under both flat and layerwise clipping styles) only affects the convergence and calibration.
- The paper observes that DP models trained with small clipping norm usually achieve the best accuracy, but are poorly calibrated and thus unreliable. In contrast, DP models trained with large clipping norm enjoy the same privacy guarantee and similar accuracy, but are significantly more calibrated.

Weakness:

1. The paper does not provide empirical validation of its theoretical results on real-world datasets, and it remains to be seen whether the observed trade-off between accuracy and calibration holds in practice.

2. The data and the NN structure in the paper are quite small. More experiments and large data (such as ImageNet) are needed

3. Did not compared with SOTA methods such as large batch.

---

> ### Author Response · Authors · 2023-04-23
> **Response to Reviewer 8Xk4**
>
> We thank the reviewer for the comment and hope our clarification will help. We are happy to include more details, e.g. number of model parameters, and to mark the accuracy-calibration trade-off more explicitly in the revision.
>
> Weakness 1: The paper does not provide empirical validation of its theoretical results on real-world datasets, and it remains to be seen whether the observed trade-off between accuracy and calibration holds in practice.
>
> **Response** We do have provided empirical validation of theoretical results, on different datasets and in different forms. To be specific, we provide the accuracy-calibration trade-off in Figure 3 with state-of-the-art DP results on CIFAR10: small R leads to accuracy=97.17% and ECE=0.9%, whereas a 10x larger R leads to accuracy=97.03% and ECE=0.7%. The same story can be seen from Figure 4 and Figure 5: small R (R=0.1) has better accuracy and better loss (Figure 4), but worse calibration/reliability diagram (right subplot of Figure 5. We also demonstrate the trade-off on MNIST (see Figure 6 right subplot) and SNLI (see Figure 8, middle & right subplots). We are very happy to point the reviewers to the empirical validation that is wanted.
>
> Weakness 2: The data and the NN structure in the paper are quite small. More experiments and large data (such as ImageNet) are needed.
>
> **Response** The data and NN structure are deliberately designed to cover small-to-large scale. For data, they range from MNIST (50,000 samples) to SNLI (550,152 samples). For NN structures, they range from CNN (17 million param) to transformers (ViT 86 million param; BERT 108 million param). We feel that it is hard for us to agree that those are small NN structures. Note that our transformers are of the same scale as the largest DP models in the current literature [1][2][3][4], and SNLI is of the same scale as ImageNet.
>
> Weakness 3: Did not compared with SOTA methods such as large batch.
>
> **Response** We would like to assure the reviewer that our experiments are consistent with SOTA, i.e. always using the same batch size as the existing literature. For instance, we use the same batch size on CIFAR10 (B=1000) as in [1], on SNLI (B=32) as in [4], on MNIST (B=256) as in [5]. The batch sizes are clearly marked in the caption of Figure 4/6/8. We emphasize that the batch sizes also cover a small-to-large scale for diversity, and it is not the case that large batch size is always the best.
>
>
> *[Summary]:* As we discussed above, we have considered large datasets and large NN structures in the original manuscript. The most important takeaway is that our theoretical claims are indeed supported by the empirical results. In principle, we agree that with more experiments, the claim could be even more convincing, yet we have good confidence that the existing experiments already provide good support to our analysis, and as a paper that also emphasize on theoretical findings, we would love to investigate more into pure empirical verifications in future work.
>
>
> [1] Bu, Zhiqi, Jialin Mao, and Shiyun Xu. "Scalable and Efficient Training of Large Convolutional Neural Networks with Differential Privacy." In Advances in Neural Information Processing Systems.
>
> [2] De, Soham, Leonard Berrada, Jamie Hayes, Samuel L. Smith, and Borja Balle. "Unlocking high-accuracy differentially private image classification through scale." arXiv preprint arXiv:2204.13650 (2022).
>
> [3] Li, Xuechen, Florian Tramer, Percy Liang, and Tatsunori Hashimoto. "Large Language Models Can Be Strong Differentially Private Learners." In International Conference on Learning Representations.
>
> [4] https://opacus.ai/tutorials/building_text_classifier
>
> [5] Papernot, Nicolas, Abhradeep Thakurta, Shuang Song, Steve Chien, and Úlfar Erlingsson. "Tempered sigmoid activations for deep learning with differential privacy." In Proceedings of the AAAI Conference on Artificial Intelligence, vol. 35, no. 10, pp. 9312-9321. 2021.

---

### Comment · Action_Editors · 2023-04-19
**paper discussion**

Dear authors,

Just a kind reminder that the discussion phase has started a week ago, and will end two weeks after the last review was posted.

From the reviews that have been posted so far, it seems that not all claims in the paper are rigorously supported. Please clarify these concerns.

-AC

---

### Comment · Action_Editors · 2023-05-23
**Clarification of Fact 5.1**

Dear authors,

Could you please clarify the statement you make in "Fact 5.1" (and also the corresponding proof)?

---

> ### Author Response · Authors · 2023-05-25
>
> Thank you for your ask. Fact 5.1 is based on the two preliminaries:
>
> (1) the privatized gradient in Equation (2.2), regardless of the choice of $R$, is equally DP.
>
> (2) Hence different optimizers like SGD and Adam produce different results that are equally DP, because they are regarded as different post-processing of the same privatized gradient.
>
> For a quick reference, we refer to (1) "For the Gaussian noise that we use, if we choose $\sigma$ in Algorithm 1 to be $\sqrt{2\log(\frac{1.25}{\delta})}/\epsilon$, then by standard arguments each
> step is (ε, δ)-differentially private" in [1] to see that the clipping norm (in their context $C$ is equivalent to our $R$) does not affect the privacy accounting, only the noise scale $\sigma$; (2) https://programming-dp.com/ch4.html#post-processing that post-processing preserves DP. We are happy to include this in the proof of Fact 5.1!
>
> [1] Abadi, Martin, Andy Chu, Ian Goodfellow, H. Brendan McMahan, Ilya Mironov, Kunal Talwar, and Li Zhang. "Deep learning with differential privacy." In Proceedings of the 2016 ACM SIGSAC conference on computer and communications security, pp. 308-318. 2016.

---

> > ### Comment · Action_Editors · 2023-05-26
> >
> > Thank you for you quick reply.
> >
> > - If the statement is meant to be a novel contribution of this work, then I still think the proof details are not satisfactory. Several notation symbols, like $\mathbf{g}$ and $B_t$, are not defined (although their meaning can be guessed), and the steps in the proof would warrant further explanations.
> > - If the statement is meant to summarize findings of other works -- which I assume is the case based on your previous answer -- then please add the proper citations in order to not misrepresent the contributions of this work.
> >
> > Furthermore, the provided reference [1] only shows (as per your verbatim citation), that for a specific choice of $\sigma$, the same privacy guarantees can be established for various $R$. Your statement in Fact 5.1 could be misinterpreted as
> > - a statement that holds for *any* arbitrary $\sigma$ (which is not the case)
> > - the formulation "are equally ($\epsilon,\delta$)-DP" is not only ambiguous, but also seems to stipulate a claim about the privacy *properties* of the algorithms, while the reference [1] only makes a claim about the privacy *guarantees* (in the sense of upper bounds on the $\epsilon$/$\delta$ parameters).

---

> > > ### Author Response · Authors · 2023-05-26
> > >
> > > Thank you for the detailed comment! The statement is not a novel contribution but a common fact in the field of DP deep learning, adopted implicitly in multiple papers (e.g. previous works have carried out ablation study on the clipping norms, without needing to adjust the privacy accounting). For instance, Figure 8(b) in "Large Language Models Can Be Strong Differentially Private Learners" (https://arxiv.org/pdf/2110.05679.pdf), the clipping norm $C$ (and other non-privacy-related hyperparameters such as learning rate) can be tuned without affecting the DP guarantee. We restate this Fact 5.1 explicitly only to assure readers that using a larger clipping norm can benefit calibration and convergence, while remaining equally DP as using a smaller clipping norm.
> > >
> > > We have added the citation in Fact 5.1 and explicitly change $\epsilon,\delta$ to $\epsilon(\sigma),\delta(\sigma)$ to make it clear that these parameters only depend on $\sigma$, not $R$ nor the type of optimizers. We would like to highlight that Fact 5.1 does hold for any arbitrary $\sigma$ in the sense that, two different optimizers, DP-SGD with $R=1$ and DP-Adam with $R=100$, running with $\sigma=1$ have the same DP guarantee, say $\epsilon=2$. DP-SGD with $R=1$ and DP-SGD with $R=100$ running with $\sigma=10$ have the same DP guarantee, say $\epsilon=0.3$, though it is different from the $\epsilon$ derived from $\sigma=1$. We are more than happy to make further modifications if the new statement still leads to misinterpretation.
> > >
> > > Additionally, $(\epsilon,\delta)$-DP is both a property and a guarantee, for example, an $(1,0.01)$-DP algorithm is also $(\epsilon,\delta)$-DP, as long as $\epsilon>1$ and/or $\delta>0.01$. Therefore, it is correct to say two algorithms are equally $(\epsilon,\delta)$-DP even though the DP characterization may be not tight. We have emphasized "privacy guarantee" not property in the first sentence of Section 5.1.

---

### Decision · Action_Editors · 2023-05-30

**Recommendation:** Accept with minor revision

**Comment:**

The reviewers commented that this paper studies an interesting problem for the privacy community and uses different techniques than previous work. However, while they overall liked the paper, they also pointed out that there are currently few practical/actionable takes from the analysis.

In light of the concerns raised by reviewer VPen, I recommend 'accept with minor revision'. Please add the clarifications as per discussion with reviewer VPen to the final version.

**Audience:**

Yes. This paper contributes towards understanding the influence of the DP training parameters, which is a topic of interest to the TMLR audience.


**Claims And Evidence:**

The paper aims at analyzing the effect of clipping and noising the gradient on the convergence of DP-SGD.
The paper investigates the effects of gradient clipping theoretically in a continuous-time perspective and complements these findings with a numerical study.

The reviewers found the evidence sufficient to support the claims. However, also note that some of the statements where previously known to the community (such as e.g. 'overclipping introduces bias').

The reviewers (and I) were concerned with a lack of rigor in certain parts or statements. These have been addressed to some extent in the discussion so far, and we are convinced that the final gaps can be bridged in the revision.